# Integrated Benefits to Agriculture with *Trichoderma* and Other Endophytic or Root-Associated Microbes

**DOI:** 10.3390/microorganisms12071409

**Published:** 2024-07-12

**Authors:** Gary E. Harman

**Affiliations:** Cornell University, Geneva, NY 14456, USA; geh3@cornell.edu

**Keywords:** bacteria, biotic and abiotic stresses, commercialization, fungi, insects, integration of biological and chemical systems, nematodes, plant-microbe interactions, photosynthesis, sustainability

## Abstract

The use of endophytic microbes is increasing in commercial agriculture. This review will begin with a strain selection. Most strains will not function well, so only a few provide adequate performance. It will also describe the endophyte–plant relationship and the fungi and bacteria involved. Their abilities to alleviate biotic (diseases and pests) and abiotic stresses (drought, salt, and flooding) to remediate pollution and increase photosynthetic capabilities will be described. Their mechanisms of action will be elucidated. These frequently result in increased plant yields. Finally, methods and practices for formulation and commercial use will be described.

## 1. Introduction

Many microorganisms are becoming important components of agricultural production systems. There are two reasons for this. First, there have been improvements in our understanding of mechanisms of action and the benefits they provide. Second, there have been improvements in formulations and in the ability to integrate biologicals into agricultural systems. Chemical pesticides and fertilizers are usually used, and so biologicals must be integrated into this chemical world.

This chapter will (A) examine the mechanisms of action, (B) describe the benefits to plants and agriculture, and (C) consider integration of biologicals, including microbe–microbe mixtures and the integration of biologicals into chemically based agriculture.

## 2. Part 1. Holobionts and Benefits to Plants and the Environment

### 2.1. Holobionts and Endophytes

Any organism does not exist in isolation, but instead is in association with numerous other organisms. This aggregation is termed a holobiont [1]. In some cases, bacteria or fungi internally colonize plant roots, and frequently, but not always, are located only there. However, some Rhizobial endophytes ascend into the upper portions of plants [2] and *Clonostacys rosea* colonize plant leaves when applied topically [3]. Some of these may have little effect on plants, be detrimental (e.g., pathogens), or may have beneficial effects. We have designated plants colonized by beneficial microbes as enhanced plant holobionts (EPHs) [4]. Some of the most useful are endophytes. Endophytes are fungi or bacteria that live within the interior of the plant and cause unapparent and asymptomatic infection [5]. The organisms persist and live for at least the life of an annual crop and are true symbionts—the plants supply nutrients and a habitat, while the microbes provide numerous benefits to the plants [6,7]. *Trichoderma* added as seed treatments grow rapidly from the seed onto the radical and then grow and become established in the interior of the root. They usually are limited to the root, and do not become established in the above-ground portions of the plant. Salicycilic acid is apparently required for localization of Trichoderma to roots [8]. Many other fungi and bacteria are endophytic, and can potentially be used for seed treatment. These include fungi in the genera *Trichoderma* [9,10,11], *Aspergillus* [12], *Penicillium* [13], *Clonostachys* [14], *Piriformaspora* [15], and *Yarrowia lipolutoca* [16], as well as bacteria in the genera *Pseudomonas* [17] and *Bacillus* [17,18] and in the family Rhizobiacae [2]. Poveda et al. [19] provides a list of more than 50 endophytic fungi.

### 2.2. Benefits to Plants and the Environment

This section will describe the advantages that microbes provide to plants, but will address mechanisms of action later. Many of the mechanisms are similar regardless of the benefits conferred. However, many endophytes cause systemic benefits to plants. For example, many of them only colonize roots, but provide benefits throughout the plant. For example, *Trichoderma* strains usually only colonize roots, but still can protect leaves from powdery mildews and anthracnose.

Many, but not all, are endophytes. For example, there are more than 100 products based on *Trichoderma* for control of diseases [10]. Nematodes are controlled by specific strains of *Trichoderma* [20]. Nematodes are controlled by specific strains of *Trichoderma* [21]. *Trichoderma* may also control nematodes through direct predation, which is facilitated via production of enzymes that degrade nematode egg shells [21]. While numerous pathogens or pests are controlled, most strains control only a few pests, so it is important to determine which products or strains control which pests. Fungi, oomycetes, and nematodes [22] are controlled by different microbes, and control of pests, including diseases, insects, and even viruses is provided by microorganisms. While numerous pathogens or pests are controlled, most strains control only a few pests, and so it is important to determine which products or strains control which pests. Fungi, oomycetes, nematodes, and insects are controlled by different microbes. Table 1 provides a summary of some of these. 

Some fungi, especially *Beauveria* and *Metarhizium*, control insects, and these fungi may be endophytic [21,22]. Surprisingly, even *Trichoderma* have been reported to have insecticidal activity [19,31]. It must be emphasized that not all members of a species or genus have activity against all of the pests listed; for example, we have screened thousands of *Trichoderma* strains and found only one that gave effective nematode control. The effective strain was an *atroviride*, but other strains of that species were ineffective. The author expects that only specific strains would be effective against insects.

However, many, if not all, pathogens and pests can be controlled by microorganisms, and the list continues to grow rapidly. Microbial pest control is becoming an integral part of agricultural practice. 

### 2.3. Strain Selection

Only some strains of any bacteria or fungus will provide adequate performance in agriculture. The author has screened thousands of *Trichoderma* strains, but only four were advanced to actual products. Strain T22 is a good example. It was originally produced via protoplast fusion [32]. Its parents were a strain with good rhizosphere competence (T95), and in some soils, it provided good protection against seed rots. However, in other soils it was ineffective. It was discovered that in soils where it was ineffective that the available iron was limited. In this situation, bacteria destroyed the hyphae due to their production of iron-chelating compounds (siderophores) [33]. The other parent (T12) was adapted to the low iron conditions, probably by producing its own siderophores. More than 100 individual progeny were produced, and T22 was one of them. Analysis of this strain suggested that its genome was primarily of the T12 genome with some segments of T95 [34]. This was recently confirmed, genetic sequence analysis showed that the genome of T22 was primarily composed to the T12 genome, but with patches of T95 [35]. This strain is widely used commercially and in research; a Biosis search revealed more than 100 studies which featured this strain.

### 2.4. Evolution of Trichoderma

*Trichoderma* strains are known for their abilities to produce extracellular enzymes and to parasitize other fungi. In its earliest forms, it was a parasite on wood-decomposing fungi, and then became a soil-inhabiting saprophyte. In the Creataceous-Paloeogene extinction event, it acquired the genes for hydrolytic genes by a massive horizontal transfer of genes from other fungi [36,37]. Thus, the events in the natural evolution of these fungi are similar to that which occurred as T22 was developed. These mycoparastic fungi are well suited to gene transfer events since the internal compartments of the prey and parasites are in close proximity. Gene transfer events are made even more likely because each cell of these fungi contain as many as 50 separate nuclei, and so gene transfer becomes likely. This is a tremendous driver of diversity and adaptation, since, as environmental conditions change, different nuclei may contain genes to exploit the new conditions [34].

### 2.5. Mitigation of Abiotic Stresses

Abiotic stresses, such as drought, salinity, flooding, and adverse temperatures lead to reductions in growth and yield of plants [38]. Several endophytes can alleviate deleterious effects of these stresses through their interactions with plants [38,39,40]. 

### 2.6. Drought 

*T. afroharzinaum* (formerly *T. harzianum*) reduced the effects of drought in greenhouse tomatoes and in field grown corn [41,42] and in wheat [41]. Mycorrhizal fungi had similar effects on watermelons [43]. An endophytic strain of *Alternaria* provides resistance to drought in tomatoes [44]. Similarly, a strain of *Curvularia* mitigated the effects of drought on rice [45]. Root colonization by a strain of *Pseudomonas chlororaphis* induced resistance to drought in *Arabisopsis* [46]; another strain of the bacterium induced drought tolerance in soybean [47]. Similarly, treatment of plants with either *Achromobacter piechaudii* or *Pseudomonas putida* reduced drought symptoms in tomato [48]. 

### 2.7. Salinity

*Trichoderma* strains alleviated effects of salt damage in tomatoes [41,49]. Similarly, *Bacillus* [50], *Piriformaspora indica* [26,51], *Dietzia* [52], and *Yarrowia* [16] all minimized the effects of salinity stress on plants. Application of *Achromobacter piechaudii* minimized effects of salinity as well drought [49]. 

### 2.8. Flooding

Endophytes can also reduce symptoms of flooding. In our trials, application of a mixture of two strains of *Trichoderma* reduced damage during flooding conditions. Many of the bacteria and fungi described in the previous two sections also alleviated flood damage.

### 2.9. Enhanced Mineral Nutrition

Arbuscular mycorrhizal fungi are obligate plant symbionts. They form cellular structures between plant cell walls and the plasmalemma. They also form extensive networks of hyphae in the soil and that grow on plant roots. These networks form within roots of the same plants and even between different plants, even between plants of different species. These networks accumulate phosphorus and other nutrients and transfer this to the plants. This enhances mineral nutrition of plants, and thereby enhances plant productivity [53].

*Trichoderma* and other endophytic organisms can also improve plant nutrient status. Some metallic nutrients are poorly soluble and are therefore unavailable to plants. *Trichoderma* strains are able to solubilize essential nutrients [53], and they also can enhance nitrogen uptake [54]. *Pseudomonas* and *Trichoderma* produce siderophores, which are compounds with high affinity for iron and other metallic plant nutrients [55,56]. 

### 2.10. Competition

Competition for space or nutrients has been suggested as a mechanism for biocontrol, but usually without good evidence. However, in one case, this is the mechanism. *Aspergillus flavus* infects cotton bolls and other crops. In cotton, it frequently colonizes wounds made in the boll by boll worms. In the boll it produces aflatoxin which is a strong carcinogen. The FDA only allows 20 ppb of the toxin (www.canr.msu.edu/news/trending-aflatoxins). Peter Cotty and his colleagues searched for naturally occurring stains of *A. flavus* and used high levels of the atoxigenic strain to inoculate fields with high levels of the atoxigenic strain. This resulted in good control of aflatoxin by competitively colonizing the infection court. Thus, the atoxigenic form functioned as an endophyte and resulted in good reductions or eliminations in aflatoxin production [57].

### 2.11. Alleviation of Environmental Pollutants

Pollution of soils by, for example, heavy metals, organic compounds, or other pollutants can adversely affect plant growth and pose health risks to humans. *Staplococcus arletta* alleviated chromate toxicity in sunflowers by suppressing the uptake of the toxicant and by reducing the hexavalent form to the less toxic trivalent form [42]. However, beneficial effects do not always occur. In a commercial planting of poinsettia, a report was received that said after application of *T. afroharzianum* all the plants died, while in its absence, the plants were healthy. It was discovered that the plants were grown in sewage sludge with a high level of chromium. *T. afroharzianum* increased the uptake of the metal to toxic levels.

The ability of a gibberellin-producing strain of *Penicillium janthinellumI* minimized the effects of aluminum toxicity in tomato. Salicylic acid was upregulated in the presence of the fungus, and cell membranes were less damaged [58].

Cyanides are frequently present in mine tailings. Endophytic strains of *Trichoderma afroharzianum* were able to take up and degrade metallocyanins [59]. Water used in olive processing became contaminated with toxic polyphenols. Strains of *Trichoderma* were able to degrade these toxic compounds [59].

Soils polluted with oily wastes from petroleum drilling or other industrial processes are serious environmental hazards. Plants colonized by *Pseudomonas putida* were able to grow in soils contaminated by oil, and the oil was eventually broken down to less toxic materials. The bacterial strains themselves probably did not break down the oil, but they permitted roots to grow by increasing plant stress tolerance and other soil organisms growing on the roots degraded the oil [60].

### 2.12. Enhanced Photosynthesis

All the benefits described require fixed carbon and energy in the form of photosynthate. This must come from photosynthesis [61]. Therefore, for the benefits to EPHs to occur photosynthetic rates need to be increased. Plants that contain endophytes frequently are greener [56]. There are anatomical changes in leaves of plants colonized by endophytes, including higher density of smaller stomata, thicker palisade parenchyma, and larger intercellular spaces in the mesophyll. The modifications in leaf functional anatomical traits affected gas exchanges [62]; in fact, starting from the reproductive phase, the rate of leaf net photosynthesis (NP) was higher in inoculated compared to control plants. These data are consistent with the better maximal photosystem II photochemical efficiency observed in inoculated plants; conversely, no difference in leaf chlorophyll content was found. The PGPM (plant-growth-promoting microbes) induced changes in leaf structure and photosynthesis, leading to an improvement of plant growth [62]. 

In plants treated with *T. asperellum*, many photosynthetic genes were up-regulated [63]. In EPHs, levels of photosynthetic pigments and proteins were increased [23,63,64]. Rhizobia also demonstrates such effects [2]. There are many reports of the up-regulation of genes encoding for photosynthetic functions, and chlorophyll concentrations were increased [65]. These are summarized in [66].

### 2.13. Enhanced Yield and Plant Growth

Many endophytic microbes enhance plant growth. Rhizobia are used around the world to supply soluble nitrogen to legumes, and this increases plant growth, especially under conditions of low nitrogen fertilizer. *Piriformasora indica* inoculation results increased plant growth and development, in part through reprogramming genes involved in various physiological processes [51]. In watermelon, mycorrhizae increased plant growth, in part through inactivation of reactive oxygen species (ROS) [42]. There are many reports of the ability of *Trichoderma* to increase plant growth, including [56], and these advantages across several endophytes has been summarized [4]. A summary of plant growth promotion by mycorrhizae, *Trichoderma*, *Bacillus*, and *Peudomonas*, together with the traits associated with them is available [66]. Evidence for induced resistance as the primary method of control is as follows:Biocontrol (and other) attributes of *Trichoderma* occur at locations distant from the fungus, such as control of leaf disease when *Trichoderma* is only in the root.The mechanism of several beneficial abilities of the fungus are similar and related to translocation of signals throughout the plant, including, for example, biocontrol and increased photosynthetic activity.With *T. virens*, mutants were made that differed in mycoparastic ability, production of the antibiotic pyoluterorin, and ability to induce systemic resistance. Deletion of mycoparasitic ability and antibiotic production had no effect on biocontrol capability, but induced resistance was essential [7,67].

### 2.14. Mechanisms

#### 2.14.1. Antibiosis

In some cases, antibiosis is part of the mechanisms of biocontrol. For example, phenazines have been implicated in the biocontrol of late blight caused by *Gaeumannomyces* [30,68]. More than 40 antibiotic compounds have been reported as being produced by *Trichoderma* [69], but other mechanisms may be more important in biocontrol. It was thought that antibiosis was critical for control of damping off by *Trichoderma* [67], but in later studies, neither antibiosis nor mycoparaitism were correlated with disease control. Instead, the ability to control the disease was tightly linked to systemic resistance [7]. However, *Trichoderma* produces substances that have antibiotic or disease-preventing activity including terpenes, non-ribobomal peptides, and ketones [10].

#### 2.14.2. Parasitism

For many years, mycoparitism was considered to be an important factor in biological control by *Trichoderma*. More recently, attention has focused on systemic resistance as a more important mechanism [7]. However, *Ampelomces quisqualis* is an obligate pathogen of powdery mildews, and this is the active ingredient of Aq10, a commercial biocontrol product [24], and *Coniothrium minitans* is a parasite of sclerotia of *Sclerotium* spp. [70]. In these two cases, parasitism is the sole mechanism of action.

#### 2.14.3. Competition

Competition for space or nutrients has been suggested as a mechanism for biocontrol, but usually without good evidence. However, in one case, this is the mechanism. *Aspergillus flavus* infects cotton bolls and other crops. In cotton, it frequently colonizes wounds made in the boll by boll worms by various genera including *Diaparopsis*, *Earias*, *Helicovepa,* and *Pectinophora)*. In the boll it produces aflatoxin which is a strong carcinogen. The FDA only allows 20 ppb of the toxin (www.canr.msu.edu/news/trending-aflatoxins) Peter Cotty and his colleagues searched for naturally occurring stains of *A. flavus* and used high levels of the atoxigenic strain to Inoculate fields with high levels of the atoxigenic strain. This resulted in good control of aflatoxin by competitively colonizing the infection court. Thus, the atoxigenic form functioned as an endophyte, and resulted in good reductions or eliminations in aflatoxin production [71].

#### 2.14.4. Antifeedants

Control of insect pests may be through the production of microbial metabolites that act as antifeedants. In some cases of insect control by *Trichoderma*, this is the mechanism [19]. Root and seed feeding dipterans detect the metabolites of roots, seeds, and volatile exudates from microbial activity on these plant parts and preferentially lay eggs in this area. The emerging larvae feed on the plants, thus severely damaging them. A seed or soil treatment with *Cheatomium globosum* alters the microbial community so that they are no longer attractive to the female dipterans and damage is lessened [72,73].

### 2.15. Systemic Reactions and Differentially Expressed Genes and Proteins

Many, and probably most, endophytes induce systemic reactions in plants. Mention has already been made of control of foliar diseases when the endophyte is located only in the root. Some examples of microbes that induce systemic resistance include *Trichoderma* [4,74], *Pseudomonas* [71], *Rhizobium* [2], *Bacillus* [68], and *Piraformaspora indica* [75].

### 2.16. Elicitors

Endophytic organisms may be located in the outer layers of the root [75,76], and produce signaling compounds that usually interact with receptors in the plasmalemma [4]. There are numerous types of such molecules, which are designated as effectors or affectors, also known as microbial associated recognition patterns. They include volatile compounds such as 6-penyl-α-pyrone and 1-octen-3-ol, and other small molecular weight molecules such as harzianic acid, heptelic acids, koningic acid, chitoologosaccharides, and glucans [4,77]. Proteins may also function as effectors; small protein 1 is produced by *T. virens,* and is essential for induced resistance in maize [78]. Peptabiols are linear peptides that induce plant defense responses [79]. Specific hydrophobins also function as effectors, and even sRNAs may play a role [80]. With *Trichoderma*, release of sRNA leads to the upregulation of plant cytoplasmic nucleotide-binding site leucine-rich receptors. After *Trichoderma* recognition, the plant genome reprograms genes involved in defense and development, and activates the antioxidant system phytohormone that leads to increased systemic resistance [36]. 

Once affectors interact with cell membranes, then systemic signals are produced that are translocated throughout the plant via map kinases [35]. Many, or perhaps most, plant symbionts alter gene regulation. For example, in rice seedlings, there were 301 transcripts that were up-regulated, and 420 were related to photosynthetic functions [62]. In *P. indica*, disease resistance requires reprogramming of genetic components [50], while in plants colonized by *Pseudomonas*, the activation of plant defense systems occur [71].

Endophytes have novel methods to induce resistance to biotic and abiotic stresses. As noted earlier, active defense systems involve a cost to the plant, since the production of defense-related compounds and systems are expensive in terms of their energy and fixed carbon requirements. Plants have evolved systems to avoid these requirements. Gene priming is a system whereby genes and proteins involved in defense reactions do not have to be continually produced. Instead of continual production of these substances, changes in the upstream regulatory portions of genes, including alterations of the cellular chromatin, by modification of histones, or by DNA methylation [81,82,83]. Consequently, in the absence of stress, the defensive materials are not produced, but when stressful conditions occur, the modifications in the upstream regulatory regions are activated quickly.

This induced resistance is designated as priming, which can be defined as “an adaptive strategy improving plant defense capacity whereby an initial stimulus activates the physiological, transcriptional, metabolomic and epigenetic mechanisms that enable the plant to respond more rapidly and/or efficiently to subsequent exposure to a biotic or abiotic stress” [36]. Since priming occurs via epigentic changes in the DNA, it can be passed to offspring, thus conferring heritable resistance [36,84].

### 2.17. Reactive Oxygen Species (ROS)

When plants (or any other organism) are under stress, deleterious substances are frequently produced. Among the most damaging are reactive oxygen species, and they inactivate proteins, damage membranes and nucleic acids. These compounds include free radicals, the superoxide anion (O_3_+), H_2_O_2_, and other highly reactive compounds. To counteract the damage caused by these compounds, plants produce antioxidant molecules. However, once the antioxidants react with ROS, they are themselves inactivated [85]. 

There are various enzymes with reductive abilities that regenerate the active form of the antioxidants [84]. Endophytes have the capability to up-regulate the genes that code for the reductive enzymes. This is a major mechanism of disease, pest, and abiotic stress resistance. Much of the damage to cellular systems is caused by ROS, and the up-regulation of these ameliorates this damage. Endophytes with this ability include *Trichoderma* [86,87], mycorrhizae [43], and *Piraformaspora indica* [88,89].

### 2.18. Acc Deaminase

ROS are not the only metabolites that damage plant cells under biotic or abiotic stresses. Ethylene, like ROS, have signaling functions in plants, but cause injury to tissues if at high levels. Acc deaminase (1-aminocyclopropane-1-carboxylate) breaks down ACC, which an immediate precursor of ethylene to ammonia and α-ketobuyrate [90]. This enzyme is produced by various bacterial root endophytes, including *Pseudomonas* spp. [91] and fungi such as *Trichoderma* [92]. When the genes encoding ACC deaminase are produced in plants, they ameliorate effects of many kinds of stress. 

Thus, a wide range of endophytic bacteria and fungi enhance plant growth through similar mechanisms, even though they are taxonomically diverse. The diverse benefits to plant growth and development are common to numerous endophytes.

## 3. Part 2. Integrated Solutions for Commercial Agriculture

The preceding sections have described the scientific underpinnings of the advantages of endophytes. However, endophytes are only useful if they are amenable to commercial agricultural practices. This section will describe practices and systems to deploy endophytes in commercial agriculture.

### 3.1. Large-Scale Production and Formulation Methods

Methods for producing large amounts of endophytes are necessary. The methods used need to be rapid and inexpensive. For bacteria liquid fermentation is widely used, while for fungi semi-solid fermentation is preferable. It is not sufficient just to produce the endophyte; the material produced must have adequate shelf life, and the physiological state of the material is important. For example, *Trichoderma* needs to be acclimated to low moisture levels, otherwise shelf life is inadequate. Adding an osmoticant such as glycerol to the growth medium extended shelf life of the fungal preparations [93].

Formulation methods are required. In many cases, the earliest formulations were powders that were applied to seeds at the time of planting. These were effective, but are too laborious for most farmers. Liquid formulations are preferred and are frequently essential for large-scale agriculture. Liquids can be either water or oil based. Fungal spores are frequently covered with hydrophobins, which are proteins with one hydrophobic and one hydrophilic end. They are small cystine-rich molecules [94]. If fungal spores are dried and mixed with oils, they form a stable suspension. If suspended in oils, they become immiscible with water. However, if suspended in water, they then become immiscible with oils. This is due to the reorientation of the hydrophobins to become either hydrophobic or hydrophilic. This property is useful in formulations to provide concentrated and effective fungal products.

Encapsulations are also useful in formulations. Coating of inoculum with materials such as glucans can frequently provide longer shelf life [95]. Similar improvements in shelf life are possible through the use of invert emulsions. Invert emulsions are mixtures of water, oils, and a surfactant (mayonnaise is a good example). They can provide enhanced shelf life, but also increase compatibility with chemical pesticides. 

### 3.2. Integrated Biological–Biological and Chemical–Biological Treatments

It seems paradoxical to consider the use of fungi with chemical fungicides, but it is possible. Almost any biological fungicide can safely be coated onto seeds if the chemical pesticide is allowed to dry before application of the biological. Also, many *Trichoderma* strains are resistant to fungicides [79]. This permits the use of biologicals in high throughput commercial systems. One method is to treat seeds with chemical liquid slurry treatments, and, upon exit from the seed treater, application of the biological via a secondary treater. It also is possible to use chemical–biological treatments using alternations of the biological and the chemical treatments. Of course, if endophytes that internally colonize plants, then the living organism does not come into contact with the pesticide. In that case, the strong but brief protection is provided by the chemical treatment, while the long-term advantage to plants is provided by the biological treatment. This combination frequently provides better plant protection against biotic abiotic stresses, and may increase crop yields. Biological–biological treatments may also be advantageous. Examples are *Rhizobium*–*Trichoderma* treatments that both improve plant nutrition and provide the other benefits just described. Other commercial products may include *Trichoderma–Trichoderma*, *Trichoderma*-*Bacillus*. In addition to seed treatments, biologicals can be applied to potting mixes as drenches when transplanting, and usually are fully compatible with any chemical that may be applied. Co-inoculation with *Trichoderma* and the AMF fungi provided disease control and/or plant yield enhancement [96,97]. Examples of *Trichoderma–Trichoderma* combinations include Sabrex-Sa (Agraauxine, https://agrauxine.com/en/) and Rootshield Plus (Bioworks, https://www.bioworks.com.au/). An example of *Trichoderma–Rhizobium* is Excaibre SA (Agrauxine). An example of a commercial product with mycorrhizae and *Trichoderma* is Great White from Plant Revolution (https://plantrevolution.com/).

## 4. Conclusions

This paper describes *Trichoderma* and other endophytes and the advantages they confer to plant agriculture. It begins with the essentiality of strain selection, since only a few gave adequate performance. *Trichoderma* as a species acquired features such as their ability to degrade cell walls through lateral gene transfers from other fungi. *T. afroharzianum* T22 became an effective strain through gene transfer. Organisms do not exist in isolation, but instead are associated with other organisms. These are holobionts. Endophytic organisms, which typically are in roots, can provide numerous benefits to plants. These include resistance to biotic and abiotic stresses and enhanced photosynthetic capabilities. They may also provide environmental benefits such as remediation of polluted lands and industrial wastes. The most common mechanism of action of most holobionts is through the activation of systemic resistance. The organisms interact with plant cell wall membranes and signals are transmitted throughout the plants to provide changes in plant gene and protein expression that provides the numerous benefits they provide. Regardless of the benefits, they cannot be used commercially unless systems of large-scale culture and formulation are developed and implemented. For commercial production liquid culturing systems are usually used, while for fungi, semi-solid production is preferred. Various formulations may be used, including powders and dusts and both water- and oil-based liquid formulations. Combinations of chemical–biological and biological–biological products and systems are available. Combinations of biologicals include *Trichoderma–Rhizobium* and *Trichoderma–Trichoderma*. Therefore, endophytic organisms are being increasingly used in commercial agriculture. This is possible because of a greater understanding of mechanisms of action, the advantage of improved strains, and the development of improved systems of production, formulation, and large-scale distribution through commercial channels.

## Figures and Tables

**Table 1 microorganisms-12-01409-t001:** Pests and pathogens controlled by different microorganisms.

Microorganism	Pest, Disease or Pathogen	References
*Trichoderma* including *asperellum*, *harzianum*, *afroharzianum*, *viride*, *atroviride*, *virens*	**Fungi and Oomycetes** *Botrtis*, *Pythium*, *Colletotrichum*, *Fusarium*, *Theiliaviposis*, *Rhizoctonia*, *Phytophthora*, *Sclerotinia*, *Didmyella*, *Chondrosteum*, *Heterobasicium*, *Alternaria*, *Didmyella*, *Verticillium*, *Ramularia*, *Macrophomina*, *Clarireedia. Cerospora* Powdery (Erysiphacea), and downy mildews (Plasmopara) and others **Viruses** Tobacco mosaic virus, green mottle mosaic virus **Bacteria** *Xanthomas*, *Pseudomas*, *Ralstonia* Nematodes Insects Leipidoptera, Hemiptera, Acari, Ortoptera, Coleoptera, Biatrodrea	[23,24]
*Clonostachys rosea*	**Fungi** *Fusarium*, *Botrytis*, *Alternaria*, *Bipolaris*, *Rhizoctonia*, *Sclerotium*, *Helmintospoium* Nematodes	[19,25]
*Ampelomyces quisqualis*	**Fungi** powdery mildew (Eryiphacea)	[24]
*Bacillus* spp, including *amyloliquifaciens*, *firmus*, *licheniformis*, *pumilus*	**Fungi** *Phytophthora*, *Fusarium*, nematodes, tomato yellow leaf curl Powdery mildew (Erysiphacea), *Puccina*, *Colletrichum*, *Rhizoctonia*, *Septoria*, *Erysiphe*, *Dreschlera*, *Ascochyta,* and others.	[24]
*Piriformaspora indica*	**Fungi and oomycetes** *Mysporella*, *Drechslers*, *Ascochya*, *Sclerotinia*, *Colletotrchlum*, *Ramularia*, *Phytophthora*, *Cercospora. Borytis*, *Alternaria* and others	[26,27]
Arbucular mycorrhizal fungi (including *Funneliformiis*, *Glomus, and Rhizophagus*)	**Fungi** *Fusarium* Nematodes	[27,28]
*Atoxigneic* *Aspergillus flavus*	**Fungi** Toxin forming *Aspergillus flavus*	[29]
*Beauveria bassiana*	**Insects** Aphids, chewing insects	[24,30]
*Pseudomonas*	**Fungi** *Verticillium*, *Fusarium*, *Blumeria*, *Cocliobolus*, *Erysihae*, *Dreschlera*, *Tilletia*, *Ustilago*, *Fusarium*, *Bipolaris*, *Microdochium*, *Botrytis*, *Phytophthora*, *Peronospora*, *Rhizoctonia*, *Botrtis*, *Rhizopu*, *Gaeumannomyces* **Bacteria** *Erwinia*	[24,30]

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
