# Peer review of "Integrated Benefits to Agriculture with *Trichoderma* and Other Endophytic or Root-Associated Microbes"

_microorganisms, 2024, doi:10.3390/microorganisms12071409_

Round 1
Reviewer 1 Report
Comments and Suggestions for Authors
Manuscript entitled “Integrated benefits to agriculture with Trichoderma and other endophytic microbes.” The manuscript summarized the benefits of plant endophytes, including Trichoderma and other microbes to agriculture. However, several points need to be addressed before it can be accepted.
The author listed quantity of microbes that involved in their benefits to plants and environment. However, several microbes in the literature were not originally endophyte, but isolated from soil, rhizosphere or other environment, although they had the ability to colonize plants. This is not according with the title of this manuscript, for the theme of this manuscript was focus on endophytes.
The part 2 section. INTEGRATED SOLUTIONS FOR COMMERCIAL AGRICULTURE. In this section, the author should supplied detailed literatures for commercial application of endophytic products to agriculture, including the current endophytic products (endophytic products, or biological-biological products, or chemical-biological products) in the market, their detailed biological effects on plants in greenhouse or field experiments (contained biocontrol ability, growth promotion capacity, or other benefits to plant).
Line 72 and 88, the section of “Strain selection” and “Evolution of Trichoderma” were under the topic of “Benefits to plants and the environment”. But the two sections had week relation with the topic. The two sections used Trichoderma T22, a wildly used commercially product as examples, compared with other related Trichoderma strains in evolution, but without the information that which type, or which characteristics of endophytes could be selected to have the potential benefit to plant and environment.
Line 46 The topic of “Benefits to plants and the environment”. I suggest construct a section of “Improve the biocontrol ability of plant” under this topic.
Table 1. Microorganism Trichoderma spp, including asperellum, harzianum, afroharzianum, viride, atroviride, virens should rewrite to “Trichoderma spp, including T. asperellum, T. harzianum, T. afroharzianum, T. viride, T. atroviride, T. virens”. They are applied to all the parts of this manuscript.
Table 1. Fungal and Oomycetes. “Botrtis, Pythium… Powdery and downy mildews and others”. This part should used pathogens of diseases, so the author should used the pathogens of Powdery and downy mildews and others here, but not the diseases. Similarly applied to other diseases of Table 1.
Comments on the Quality of English LanguageMinor editing of English language required
Reviewer 2 Report
Comments and Suggestions for Authors
When reading the review, one often gets the impression that the author raises interesting questions about certain biological properties of Trichoderma, but suggests finding the answers on one's own. However, this feature emphasizes the specialness of this review, which sets it apart from many others.
For example, the paragraph “antibiosis”. The main message of the author is that antibiotics produced by Trichoderma are not as important for biocontrol of diseases as other mechanisms. This is an interesting statement and could have been disclosed in more depth.
Or another example (Lines 66-68), what such properties might determine the efficacy of Trichoderma against insects? More detailed information would be helpful.
One more, (Lines 135-145) paragraph “Competition” - do you think this results obtained by Peter Cotty can be transferred from Aspergillus flavus to Trichoderma ?
Regarding the section “Integrated Biological and Chemical-Biological Treatments”:
There are studies that aim to improve the biology of endophytes, making them more resistant to UV radiation (DOI: 10.1016/j.jip.2011.10.004), fungicides (DOI: 10.1016/s0022-2011(02)00151-9) and so on.
In what ways might the phenotypic traits and overall life strategies of these modified endophytes be affected? For example, a reduction in the ability to colonize plants. Do you think this is possible?
--
Table 1. Which microbes listed are endophytes? they all? I think this table could use some interior lines to visually differentiate the microbes from each other.
L. 67. atrovide – species ?
L.77 and is some soils – misprint ?
L.133 Pseudomods – misprint ?
L.208. Terpenes spelled twice
Reviewer 3 Report
Comments and Suggestions for Authors
Dear Author,
Starting with reading the abstract, I found the contribution of the paper not too promising. However, after going deeper into the paper, I believe that the manuscript with minor changes must be published.
The manuscript summarizes many key considerations and previous research conducted that clearly aligns with future trends in agriculture and food production, especially when it comes to policies to reduce pesticide and fertilizers use. The structure of the manuscript is correct and the aims are justified. However, I believe that there is still room to improve the work, and I would suggest that it be accepted after minor changes and additions.
The attached review paper did not fully harmonize the aims defined in the introduction through the structure of the manuscript, nor were the quality conclusions well created. References and Latin names of species have shortcomings, so I will mention only a few, and I invite the author to show the progress of the manuscript in that segment of the work in the next submission. I will write to you about the arguments why I think so.
Abstract
Expand. What would be methodology in a classical manuscript is missing. What the author did, how he structured the manuscript, he stated the three (3) objectives nicely in the introduction, I think he should state them in the summary. They were giving an overview of the research, right? The aims should be more clearly structured and aligned with the one in the manuscript itself.
Keywords – Missing more.
Line 12 – add more keywords, I suggest: sustainable plant cultivation, plant resistance, microbiological rhizosphere enrichment. Arrange them in alphabetical order;
1. Introduction
The structure is fine. Technically, some moments need to be improved.
Line 52 – specify the reference;
Line 55 - in nematodes and in table 1. I suggest the author to consider the work of the authors: Gašparović Pinto, Ana; Kos, Tomislav; Puškarić, Josipa; Vrandečić', Karolina; Benković - Lačić, Teuta; Brmež, Mirjana Soil Ecosystem Functioning through Interactions of Nematodes and Fungi Trichoderma sp. // Publications / MDPI, 16 (2024), 7; 2885, 14. doi: https://doi.org/10.3390/su16072885 for improving the manuscript.
Table 1 has serious shortcomings, the first is the font, then the visibility and connection of references with key words in the table for which these studies were conducted. The connection between the reference and the organism must be clearer. Shouldn't references be written in the list of references in the order in which they appear, so after 21, 22 should follow, not 52 or 97, regardless of whether the table is a separate segment of the manuscript.
Line 105 – Latin name;
Line 121 – support the statements with reference;
Line 137, 148, 154, 190, - I suggest that the sentence does not start with the Latin name of the species but with the word "Species";
Line 139 – "Boll vormes" – Latin name;
Line 140 – convert website to reference;
Line 156 – Check species;
Line 178, 180 and 336 – the abbreviations PSII, PGPM and AMF, I think that even though one knows what it is about, the full name should be written for the first time in the text of the manuscript;
Line 205, 214, 228, 236, – Technical– too large a space between words;
Line 208 – What are the reasons that terpenes are mentioned twice in the sentence;
Line 221 – Need to support with a reference;
Line 249 – Sentence style to consider;
Line 270 – Match the chemical symbol O3+ and support the sentence with a reference;
Lines 313, 318 – Allegations should be substantiated with a reference;
Line 319 – Support the entire text with techniques or methods of implementing integrated solutions, examples of other authors' research and references.
Following the structure of the aims, the aim (C) consider integration of biologicals, including microbe-microbe mixtures and integration of biologicals into chemically based agriculture, should be formulated as a separate chapter, then organize the conclusions chapter.
Conclusion
I suggest that a chapter on conclusions should be created in the manuscript. There are a lot of doubts. The previous part of the manuscript is written in the style of a discussion and is not conclusive. In the last chapter of the manuscript, it is discussed and not concluded. I understand that previously everything should lead to what the author writes in the last chapter, but still it does not conclude but discusses (incompletely).
References
There are a lot of errors and shortcomings of a technical nature in terms of inconsistency in specifying the order in which references appear in the text, and their listing in the list of references (some need to be revised, and one is missing).
References: 22, 44, 47, 49, 50, 51, 54, 64, 66, 69, 76, 83, 88, 90, 91, 94,
References must be written in accordance with the rules of the journal.
In conclusion, I believe that the work needs to be further improved: reformulate the abstract, list key words, add conclusions, edit the Latin names of organisms and references in the manuscript.
The manuscript must be returned for revision.
With respect.
Author Response
Reviewer 3
Abstract
I have included a summary.
Key words
I have added more.
Section on competition
I have converted the on-line ref to a standard ref.
I have added genera of boll worms.
Refs
I have checked the refs, made corrections and I formatted the refs according to MDPI Endnote style.
The refs are listed sequentially as they appear in the Ms. I some cases the refs in the table ere cited earlier in the paper so they are not sequential in the table.

Round 2
Reviewer 1 Report
Comments and Suggestions for Authors
Accept in present form
Comments on the Quality of English LanguageMinor editing of English language required